# Effect of Gap Length and Partition Thickness on Thermal Boundary Layer in Thermal Convection

**DOI:** 10.3390/e25020386

**Published:** 2023-02-20

**Authors:** Zhengyu Wang, Huilin Tong, Zhengdao Wang, Hui Yang, Yikun Wei, Yuehong Qian

**Affiliations:** 1National-Provincial Joint Engineering Laboratory for Fluid Transmission System Technology, School of Mechanical Engineering, Zhejiang Sci-Tech University, Hangzhou 310018, China; 2School of Mathematical Science, Soochow University, Suzhou 215006, China; 3College of Mathematics and Computer Science, Zhejiang Normal University, Jinhua 321004, China

**Keywords:** thermal convection, thermal boundary layer, partition wall, gap length

## Abstract

Two-dimensional direct numerical simulations of partitioned thermal convection are performed using the thermal lattice Boltzmann method for the Rayleigh number (*Ra*) of 10^9^ and the Prandtl number (*Pr*) of 7.02 (water). The influence of the partition walls on the thermal boundary layer is mainly focused on. Moreover, to better describe the spatially nonuniform thermal boundary layer, the definition of the thermal boundary layer is extended. The numerical simulation results show that the gap length significantly affects the thermal boundary layer and Nusselt number (*Nu*). The gap length and partition wall thickness have a coupled effect on the thermal boundary layer and the heat flux. Based on the shape of the thermal boundary layer distribution, two different heat transfer models are identified at different gap lengths. This study provides a basis for improving the understanding of the effect of partitions on the thermal boundary layer in thermal convection.

## 1. Introduction

In many natural phenomena and engineering applications, flow driven by temperature difference, called thermal convection, is ubiquitously observed [1,2]. A classical simplified physical model of a fluid sandwiched by two parallel plates, with the bottom and top plates functioning as heating and cooling plates, respectively, is known as Rayleigh–Bénard convection (RBC) [3,4,5]. A key problem in RBC research is the measurement of the heat flux of the system. This heat flux can be determined using the Nusselt number *Nu*, which is defined as [6]:(1)$$Nu=1+\frac{\langle u_{y}\theta \rangle }{\left(\kappa \Delta \theta H\right)}$$
where *u*_y_ is the vertical velocity, Δ*θ* is the temperature difference between the heating and cooling plates, *H* is the fluid layer height, and < > represents the spatial average of the whole fluid domain. It depends on the motion of the convection and the physical properties of the fluid. The two main governing dimensionless parameters are the Rayleigh number (*Ra*) and Prandtl number (*Pr*):(2)$$Ra=\frac{\beta \Delta \theta gH^{3}}{\nu \kappa }\;\mathrm{and}\;Pr=\frac{\nu }{\kappa },$$
where *g* is the gravitational acceleration, and *κ*, *β*, and *ν* represent the thermal diffusivity, thermal expansion, and kinematic viscosity, respectively. Numerous RBC studies have shown that determining the relation among *Nu*, *Ra*, and *Pr* is essential. Many experimental and numerical results have also shown that *Nu* can be determined by determining *Ra* and *Pr* [7,8,9,10].

Recently, many non-traditional RBC systems have also been widely investigated to explore the flow mechanisms in different domains. For example, Wang and Zhou et al. applied horizontal vibration to investigate the effect of periodic vibration on the flow and heat transfer characteristics of the RBC system. They found that the application of horizontal vibration caused instability of the thermal boundary layer, which improved the heat flux [11]. The rotational effect has been widely studied in astrophysics and geophysics, with many scholars exploring the effect of rotational effects on turbulent heat transfer by rotating thermal convection systems [12,13,14]. Sajjadi et al. [15,16] investigated natural convection in magnetohydrodynamics by adding a magnetic field in the RBC system; it was found that increasing the Hartmann number decreases the heat transfer rate. Liu et al. [17] studied the influence of layer thickness and fluid properties on heat transfer of an RBC two-layer system through two immiscible fluid layers. They found that heat transport is dependent on layer thickness in the two-layer system. Notably, the insertion of partition walls in the RBC system forms the partitioned heat convection, which enhances the heat flux transfer [18,19,20,21,22]. Similar geometric structures exist in the microchannel heat sink. The width and height of the microchannel will affect the heat flux in the microchannel heat sink [23,24], which geometric parameters also exist in the partitioned thermal convection. Bao et al. (2015) [18] found that the insertion of partition walls in RBC systems can significantly enhance heat flux. They determined that the heat flux of the partitioned thermal convection is the highest when the gap length (the gap between the inserted walls and top/bottom boundaries) and the thermal boundary layer thickness are the same. Lin et al. [20] further investigated the relationship between gap length and thermal boundary layer thickness and observed a strong coupling between the gap length and thermal boundary layer. In thermal convection, two types of boundary layers are present near the cooling and heating plates: the kinematic and thermal boundary layers. Moreover, the heat flux and thermal boundary layer are closely related. The cold/hot plumes are generated from the top/bottom thermal boundary layer, where the fluid separates from the thermal boundary layer due to buoyancy and moves up/down with varying energy [25,26]. Therefore, the study of the effect of partition walls on the thermal boundary layer is very important for exploring the heat transfer in partitioned thermal convection.

Recently, the exploration process of partitioned thermal convection mainly focuses on the optimal gap, flow mechanism, and heat transfer characteristics; they are all closely related with the thermal boundary thickness. The study aims to reveal the influence of partition walls on the thermal boundary layer in partitioned thermal convection. We extend the definition of thermal boundary layer thickness to make it applicable to partitioned thermal convection and show the effect of partitioning on the thermal boundary layer properties. Our numerical simulation employs water as the working fluid (*Pr* = 7.02). The fluid flow and heat transfer in partitioned thermal convection with different gap lengths and partition wall thicknesses are simulated using the lattice Boltzmann method (LBM). The remainder of the paper is organized as follows. Section 2 introduces the governing equations, the geometry configurations of the computational domain, and the detailed numerical methods. Section 3 presents the results of the numerical simulation and the detailed analysis of the thermal boundary layer. Finally, Section 4 summarizes our findings.

## 2. Numerical Method

### 2.1. Governing Equations

The partitioned thermal convection system is constructed by inserting partition walls in the thermal convection. It is governed by the Oberbeck–Boussinesq approximation equations, which describe the incompressible velocity and temperature fields via the following expressions [27]:(3)∇⋅u=0,
(4)$$\frac{\partial u}{\partial t}+\nabla \cdot \left(uu\right)=-\nabla p+v\nabla^{2}u-g\beta \Delta \theta ,$$
(5)$$\frac{\partial \theta }{\partial t}+\nabla \cdot \left(u\theta \right)=\kappa \nabla^{2}\theta ,$$
where ***u*** represents the macroscopic velocity, and *p* is the fluid pressure.

The traditional methods for solving the classical Oberbeck–Boussinesq equations are the finite element method, the finite difference method, and the finite volume method, etc. Among them, LBM is a mesoscopic simulation-scale-based computational fluid dynamics method. It has the advantages of excellent parallelism [28,29], easy setting of complex boundaries, and low computational resources. LBM is widely considered an effective method for dealing with fluid motion and engineering problems [30,31]. Partitioned thermal convection can be described by the double distribution LBM. Herein, the double-distributed LBM is used to simulate the internal flow and heat transfer in partitioned thermal convection, and the local grid refinement method is used to obtain detailed information on the boundary layer region. The double distribution LBM is introduced in the following subsection.

### 2.2. Lattice Boltzmann Method

The double distribution LBM includes the velocity and temperature distribution functions. The evolution equation of the flow field is [32]:(6)$$f_{i}(x+c_{i}\delta_{t},t+\delta_{t})=f_{i}(x,t)+\left[f_{i}^{eq}(x,t)-f_{i}(x,t)\right]/\tau_{\nu }+\delta_{t}F_{i},$$
where $f_{i}$ is the density distribution function, $f_{i}^{eq}$ is the equilibrium distribution function, $F_{i}$ represents the discrete force term, $c_{i}$ is the *i* the discretized velocity vector, x represents the particle position, $\delta_{t}$ denotes discrete time steps, and $\tau_{\nu }$ is the relaxation time of $f_{i}$. The equilibrium distribution function of *f_i_* can be expressed as:(7)$$f_{i}^{eq}=\rho w_{i}\left[1+\frac{c_{i}\cdot u}{c_{s}^{2}}+\frac{{\left(c_{i}\cdot u\right)}^{2}}{2c_{s}^{4}}-\frac{u^{2}}{2c_{s}^{2}}\right],$$
where $w_{i}$ represents the weight coefficients and $c_{s}$ is the sound speed. The D2Q9 model is defined as [33]:(8)$$c_{i}=\frac{\delta x}{\delta t}\left[\begin{array}{ccccccccc} 0 & 1 & 0 & -1 & 0 & 1 & -1 & -1 & 1\\ 0 & 0 & 1 & 0 & -1 & 1 & 1 & -1 & -1 \end{array}\right],$$
(9)$$c_{s}=\frac{1}{\sqrt{3}}\frac{\delta x}{\delta t},w_{i}=\left\{ \begin{array}{l} \frac{4}{9},i=0,\\ \frac{1}{9},i=1\sim 4,\\ \frac{1}{36},i=5\sim 8. \end{array}\right..$$
where $\delta_{x}$ denotes discrete space steps. The relation between the relaxation time ($\tau_{\nu }$) and the kinematic viscosity (ν) is [34]:(10)$$\nu =c_{s}^{2}\left(\tau_{\nu }-\frac{1}{2}\right)\delta_{t},$$
where the macroscopic density and velocity can be calculated as follows:(11)$$\rho =\sum_{i}f_{i},\;\rho u=\sum_{i}c_{i}f_{i}.$$

The evolution equation of the temperature field is given as [6]:(12)$$g_{i}(x+c_{i}\delta_{t},t+\delta_{t})=g_{i}(x,t)+\left[g_{i}^{eq}(x,t)-g_{i}(x,t)\right]/\tau_{\theta },$$
where $g_{i}$ is the density distribution function of the temperature field, $g_{i}^{eq}$ denotes the equilibrium distribution function of the temperature field, and $\tau_{\theta }$ represents the relaxation time of $g_{i}$. The equilibrium distribution function of the temperature field can be expressed as:(13)$$g_{i}^{eq}=\rho \theta w_{i}\left[1+\frac{c_{i}\cdot u}{c_{s}^{2}}+\frac{{\left(c_{i}\cdot u\right)}^{2}}{2c_{s}^{4}}-\frac{u^{2}}{2c_{s}^{2}}\right].$$

The relation between the relaxation time ($\tau_{\theta }$) and the thermal diffusivity (κ) is
(14)$$\kappa =c_{s}^{2}\left(\tau_{\theta }-\frac{1}{2}\right)\delta_{t},$$
where the macroscopic temperature can be calculated using
(15)$$\theta =\frac{1}{\rho }\sum_{i}g_{i}.$$

Using the Chapman–Enskog expansion, the classical governing equations can be obtained from the evolution equation of the flow field and temperature field under the approximation of incompressible flow [35].

### 2.3. Boundary and Initial Conditions

Figure 1 illustrates the computational model of thermal convection, wherein the height of the computational model is *H*, the length of the computational domain is *L*, and the aspect ratio is Γ=L/H=1. The cooling and heating plates are placed at the top and bottom of the model, respectively. The temperatures of the cooling and heating plates are −1 and 1 (dimensionless temperature), respectively. The plates have isothermal boundary conditions. The idea of the isothermal boundary conditions approach can be expressed as [36]: (16)$$f_{\bar{i}}\left(x,t+\delta t\right)=f_{i}^{+}\left(x,t\right),$$
(17)$$g_{\bar{i}}\left(x,t+\delta t\right)=2\rho \theta w_{i}-g_{i}^{+}\left(x,t\right),$$
where $c_{\bar{i}}$ is the inverse direction of velocity $c_{i}$ (i.e., $c_{\bar{i}}=-c_{i}$). The superscript ^+^ represents the density distribution function after the collision step and before the streaming step. To simplify the computational model, the periodic boundary is considered in the partitioned thermal convection. It can be expressed as [32]:(18)$$f_{i}(x,t)=f_{i}^{+}(x+L,t),$$
(19)$$g_{i}(x,t)=g_{i}^{+}(x+L,t).$$

To form the partitioned thermal convection computational domain, four adiabatic partitioned walls of thickness *S* are inserted into the above thermal convection model. The gap length between the cooling and heating plates is *D*. The width between two adjacent partitioned walls is *W*. The above computational parameters are dimensionless, and the height of the computation domain is used to facilitate the understanding of the physical model. The dimensionless wall thickness is $S^{*}=S/H$, the dimensionless distance between adjacent walls is $W^{*}=W/H$, and the dimensionless gap length is $D^{*}=D/H$.

### 2.4. Grid Independence Test

The partitioned thermal convection model is a novel geometric structure of the RBC system. A numerical simulation of the RBC system was performed with $Ra=10^{8}$ and Pr=4.3 for unit aspect ratio to verify the accuracy of the model and code. In the RBC system, the isothermal and adiabatic boundary conditions are applied to the top/bottom and left/right boundaries, respectively. To validate the correctness of the implemented method, the *Nu* of the above RBC system simulation result was compared with that of previous studies. In Table 1, the numerical simulation results show that *Nu* = 25.78, which is consistent with the results of Bao et al. and Zhou et al. [18,37]. So, it is reliable to use our model and code to perform numerical simulations of the novel geometric structure of the RBC system.

The application of the appropriate grid resolution can save computing resources and ensure the accuracy of numerical simulation results. To select the appropriate grid resolution, the grid independence test is performed for grid resolutions ranging from 600 × 600 to 1600 × 1600. Figure 2 shows the numerical simulation results of the average *Nu* at different resolutions. This average *Nu* is obtained over the entire space and time. The tested grid resolutions are 600 × 600, 700 × 700, 800 × 800, 900 × 900, 1000 × 1000, 1200 × 1200, and 1600 × 1600, and the test results of the corresponding grid resolution are represented by box symbols in Figure 2. Figure 2 clearly shows that the average *Nu* is unstable when the grid resolution is less than 1200 × 1200. When the grid resolution is greater than 1200 × 1200, there is no significant improvement in computation accuracy with the increased grid resolution. Thus, the grid resolution of 1200 × 1200 is the optimal grid, and this grid will be used in the subsequent computations.

### 2.5. Local Grid Refinement

The fluid flow and heat transfer in the boundary region are very complex; thus, the thickness of the thermal boundary layer cannot be calculated with a single coarse grid. Herein, grid refinement is used for local regions to obtain the accurate thermal boundary layer thickness. High-resolution grids are applied to the boundary layer region, and low-resolution grids are used for the other regions. The local grid refinement method was first applied to the LBM field by Filippova et al. [38], and then, many scholars further investigated this method [39,40,41]. This method not only yields the fluid flow and heat transfer information in the region of drastic flow but also prevents the wastage of computational resources. The entire flow field is covered by coarse grid points. In the local region, the fine grid points are inserted via the grid refinement method to form a fine grid region. The coarse and fine grid regions calculate the same evolution equations but with different time steps ($\delta_{t}$) and relaxation parameters ($\tau_{\nu }$). The relation is as follows [38]:(20)$$\delta_{t}^{f}=\frac{1}{n}\delta_{t}^{c},\tau_{\nu }{}^{f}=\frac{1}{2}+n(\tau_{\nu }{}^{c}-\frac{1}{2}),$$
where *n* (*n* = 2, 4, 8…) represents the ratio of the coarse grid discrete steps $\delta_{x}^{c}$ to the fine grid discrete steps $\delta_{x}^{f}$. In the study, the local grid refinement with *n* = 4 is applied to the upper/bottom boundary regions.

## 3. Results and Discussion

In this section, the definition of thermal boundary layer thickness is extended to make it more applicable to partitioned thermal convection. The temperature field streamlines distribution, and thermal boundary thickness is mainly discussed, revealing the effect of gap length and partition wall thickness on the thermal boundary layer.

### 3.1. Temperature Field and Streamline Distribution of Partitioned Thermal Convection

Figure 3 displays the global and local time-averaged temperature fields and streamline distribution in partitioned thermal convection. The numerical simulation results are obtained with $Ra=10^{9},Pr=7.02,D^{*}=0.01,S^{*}=0.01$, and $W^{*}=0.24$. The temperature field is indicated by the color bar, with red representing high temperature and blue representing low temperature. The fluid motion is denoted by the streamlines and arrow directions. As shown in Figure 3a,b, in the cold channel (x=0.13−0.37), the fluid moves from the top to the bottom due to the density difference. In the hot channel (x=0.38−0.62), the fluid moves from the bottom to the top and carries energy due to the temperature difference. In the channel, the fluid moves unidirectionally, and cold and hot channels are alternately distributed on either side of the partition wall. The fluid exchanges heat and mass through the gap between the bottom and top on both sides of the partition wall. Figure 3a shows that the temperature gradient is larger in the top and bottom boundary regions. This thin layer with drastic temperature changes is the thermal boundary layer. To more clearly observe the thermal boundary layer, the red box region of Figure 3a is enlarged and shown in Figure 3c. The figure clearly shows that the thermal boundary layer is inconsistent at different positions. It is thinner near the gap and thicker in the region away from the gap. This indicates that the existence of the spatial nonuniformity of the thermal boundary layer and the inserted partitioned walls will compress the thermal boundary layer thickness in the gap region. The spatial nonuniformity of the boundary layer was first proposed by Werne [42], and subsequent studies showed that this spatial nonuniformity is related to the shear and horizontal position [43,44]. 

In Figure 3b, the streamline distribution accurately describes the fluid motion. In the top/bottom boundary regions, the cold/hot fluid is separated from the thermal boundary layer and moves down/up. In the gap regions, the fluid flows through the gap to achieve heat and mass exchange. To observe the fluid motion in the gap regions, the red box region of Figure 3b is enlarged and shown in Figure 3d. In the partitioned thermal convection system, the hot and cold channels are separated by inserting partition walls. Figure 3d clearly shows that the cold fluid moves downward in the cold channel. At the bottom boundary, the cold fluid separates and moves horizontally through the gap into the hot channel. The horizontal motion compresses the thermal boundary layer thickness due to the shearing effect in the gap region. This further illustrates the spatial nonuniformity of the thermal boundary layer. In the hot channel, the hot fluid separates from the thermal boundary layer and mixes with the cold fluid flowing from the gap to form a temperature difference jet. The gap between the partition walls and the top/bottom boundaries significantly reduces the thermal boundary layer thickness. The temperature difference between the top and bottom boundary regions increased with decreasing thermal boundary layer thickness, which subsequently increased the heat flux. The heat flux of partitioned thermal convection improved with decreasing thermal boundary layer thickness. 

Figure 4 depicts the time-averaged viscous entropy generation rates and thermal entropy generation rate fields corresponding to Figure 3. The viscous and thermal entropy generation rates can be calculated as follows, respectively [32]:(21)$$S_{u}\left(x,t\right)=\frac{\nu }{\theta }\left\{2\left[{\left(\frac{\partial u}{\partial x}\right)}^{2}+{\left(\frac{\partial v}{\partial y}\right)}^{2}\right]+{\left(\frac{\partial u}{\partial y}+\frac{\partial v}{\partial x}\right)}^{2}\right\},S_{\theta }\left(x,t\right)=\frac{\kappa }{\theta^{2}}\left[{\left(\frac{\partial \theta }{\partial x}\right)}^{2}+{\left(\frac{\partial \theta }{\partial y}\right)}^{2}\right]$$

In Figure 4a, the viscous entropy generation is mainly in the gap inlet region of the channel. The viscous entropy generation is small in the central channel region. This entropy generation of viscous is caused by fluid movement in the gap region, which indicates that the viscous flow loss mainly occurs in the gap region. Compared with Figure 4b, the thermal entropy generation is much greater than the viscous entropy generation. This indicates that the thermal entropy generation is dominated in partitioned thermal convection. By comparing the temperature field of Figure 3, we can clearly see that the higher thermal entropy generation rates mainly dominates in the region of high-temperature gradient. The temperature gradient is large in the thermal boundary layer region. The thermal boundary layer property is important to deeply understand the mechanism of heat transfer process. Next, the effect of partitions on the thermal boundary layer in thermal convection is discussed.

### 3.2. Thickness of Thermal Boundary Layer

Since the partitioned thermal convection system is strictly controlled by the Oberbeck–Boussinesq equation, the top and bottom boundaries are symmetric. Herein, the temperature distribution near the bottom boundary is focused on. To study the influence of partitioned thermal convection on the thermal boundary layer properties, the non-dimensional temperature Θ(y,t) is introduced:(22)$$\Theta (y,t)=\frac{\theta^{bottom}-\theta (y,t)}{\Delta \theta }$$
where $\theta^{bottom}$ is the temperature of the bottom boundary, and Δθ is the temperature difference between the bottom and top boundaries. Moreover, Θ(y)=1 and Θ(0)=0 represent the temperatures of the top and bottom boundaries, respectively. 

The horizontally averaged temperature profiles of the bulk regions in the cold and hot channels are plotted in Figure 5a,b, respectively. The temperature profiles of the cold and hot channels rapidly increase from Θ(0)=0 to the mean and maximum temperature in the bottom boundary layer region, respectively. The temperature distribution remains stable after reaching the mean temperature in the temperature profile of the cold channel. In the temperature profile of the hot channel, the temperature distribution gradually decreases to the mean temperature after reaching the local maximum. The different trends of the temperature profile in the two regions denote the spatial nonuniformity of the thermal boundary layer. To clearly observe the temperature profile in the bottom boundary regions of the cold and hot channels, the region close to the bottom boundary is locally magnified. As shown in Figure 5c,d, both profiles are clearly linear near the bottom boundary, which is similar to previous studies on thermal convection [25,44,45,46]. Zhou et al. studied the thermal boundary layer thickness as the distance between the extrapolation of the linear part of the temperature profile and the horizontal line passing through the mean temperature [25]. The temperature profiles of thermal convection and partitioned thermal convection exhibit obvious differences. The temperature profiles of the cold channel and RBC are similar, but those of the hot channel and RBC are different. The thermal boundary thickness in the hot channel cannot be accurately determined if the thermal boundary thickness definition of Zhou et al. [25] is used. Thus, in partitioned thermal convection, the thermal boundary layer thickness is defined as the distance from the profile at which the extrapolation of the linear part of the temperature profile to the horizontal line passes through the maximum local temperature. The arrows in Figure 5c,d specifically illustrate how to obtain the thermal boundary layer thickness of the cold and hot channels, respectively.

Figure 6 shows the time evolution of the thermal boundary layer thickness at different positions (x=0.25 and x=0.50) and different bulk regions (including cold channel and hot channel regions). To quantitatively describe the time evolution, dimensionless time *τ* ($\tau =t/\sqrt{H/g\beta \Delta \theta }$) is used. Figure 6a clearly shows that the time evolutions of the thermal boundary layer thickness are different at x=0.25 and x=0.50. This time evolution further illustrates the spatial nonuniformity of the thermal boundary layer. Additionally, the average thickness of the thermal boundary layer in the bulk region is calculated. Figure 6b shows that the average thickness of the thermal boundary layer in the cold and hot regions is consistent for the time evolution. The above phenomena indicate that the thermal boundary layer thickness is consistent in the average bulk region of the cold and hot channels. The spatial nonuniformity is reflected in different positions. Thus, the thermal boundary layer thickness at different positions is dynamically distributed in the cold and hot channels. If the thermal boundary layer at some position is compressed, then some parts of the thermal boundary layer must be stretched. The thermal boundary layer thickness distribution differs due to the insertion of partitioned walls. The influence of gap length and partition wall thickness on the thermal boundary layer distribution is analyzed in the following subsection.

### 3.3. Effect of Gap Length on Thermal Boundary Layer

To study the influence of gap length on the thermal boundary layer, numerical simulations are performed with different gap lengths. The time-averaged temperature and streamline distribution of the numerical simulation results are shown in Figure 7. Figure 7a,b shows that the fluid moves down/up and forms multiple vortices in the cold/hot channel. These vortices inhibit fluid motion and heat transfer. In the top and bottom boundary regions, a small amount of fluid enters the adjacent channels through the gap. The flow in the channel is complex and unstable. When the gap length is increased ($D^{*}=0.015$), Figure 7c,d shows that the flow trend of the fluid in the channel gradually becomes stable and unidirectional. The cold/hot fluid flowing from the gap mixes with the fluid separated from the thermal boundary layer, yielding a downward/upward temperature difference jet. When the gap length is further increased ($D^{*}=0.030$), Figure 7e,f shows that the mixing of the cold/hot fluid flowing from the gap with the fluid separated from the thermal boundary layer further increases. The temperature near the cooling and heating plates is directly controlled by the thermal boundary layer. The fluctuation of the high-temperature part in the bottom boundary region is closely related to the change of the thermal boundary layer thickness. To obtain the influence of the gap length on the temperature boundary layer thickness in detail, the thickness of the temperature boundary layer at each position is calculated and plotted in Figure 8. 

Figure 8 displays the time-averaged thermal boundary layer thickness at each position (x=0.12~0.62) with different *D*^*^. The figure clearly shows that the minimum of the thermal boundary layer thickness is present in the gap region. The thermal boundary layer thickness gradually increases away from the gap region. This indicates that the thermal boundary layer is squeezed when the fluid flows through the gap. Thus, in the gap region, the thermal boundary layer thickness decreases and the heat flux increases. Furthermore, the thermal boundary layer thickness in the cold channel (x=0.13−0.37) gradually increases with the gap length. The distribution of the thermal boundary layer thickness in the hot channel is irregular. According to the shape of the distribution, this irregular distribution can be divided into two types: m-type and n-type distributions. The thermal boundary layer distribution is m-type when $D^{*}=0.0050$ and $D^{*}=0.0100$ and n-type when $D^{*}\geq 0.0150$. The trend of the thermal boundary layer thickness in the hot channel with increasing gap length is not obvious. Thus, the trend of the thermal boundary layer thickness in different bulk regions does not reveal the overall influence of the gap length on partitioned thermal convection. Subsequently, the average thickness of the thermal boundary layer in time and space is calculated. 

The statistical results of $\bar{\delta_{th}}$ (average thickness of thermal boundary layer in time and space) with varying gap lengths are shown in Figure 9. The figure shows that $\bar{\delta_{th}}$ increases with the gap length. The heat transfer is closely related to the thermal boundary layer thickness. Previous research studies showed that the thermal boundary layer thickness is inversely proportional to *Nu* [7,8,9]. To further investigate the effect of gap length on $\bar{\delta_{th}}$ and *Nu*, their relation is plotted in the subgraph of Figure 9. Here, *Nu*_(0)_ and $\bar{\delta_{th}}$ represent the numerical simulation results of thermal convection without the partition walls. The figure shows that *Nu* gradually increases with decreasing gap length. The maximum value of *Nu* is obtained at *D^*^* = 0.0050. The heat flux with partition walls is approximately 250% higher than that without partition walls. Comparison of *Nu* for *D*^*^ = 0.005 and *D*^*^ = 0.03 shows that the heat flux significantly increases with the gap length. The relation between the thermal boundary layer and *Nu* is consistent with the results of Grossmann et al. and is as follows [7,8,9]:(23)$$\overset{-}{\delta_{th}}\approx \frac{H}{2Nu}$$

.

### 3.4. Effect of Partition Wall Thickness on Thermal Boundary Layer

The above numerical simulation results show that the thermal boundary layer distribution within the hot channel with varying gap lengths can be divided into two types. Thus, numerical simulations are performed with $Ra=1\times 10^{9}$ and Pr=7.02 to determine the effect of different wall thicknesses (*D*^*^ = 0.010 or 0.020) on the thermal boundary layer. Figure 10 displays the time-averaged thermal boundary layer thickness for different wall thicknesses (*D*^*^ = 0.010) at each position. The thermal boundary layer distribution within the hot channel is not affected by the increase in the partition wall thickness. The numerical simulation results for different partition wall thicknesses show that the thermal boundary layer thickness within the hot channel is m-type. There is no significant effect with the increase of the partition wall thickness for thermal boundary layer distribution. 

Figure 11 shows the effect of partition wall thickness on $\bar{\delta_{th}}$. The figure shows that $\bar{\delta_{th}}$ decreases with increasing partition wall thickness. Comparison of the thermal boundary layer thickness for *S*^*^ = 0.01 and *S*^*^ = 0.05 shows that the decrease in thermal boundary layer thickness is not significant. The subfigure of Figure 11 displays the functional relation between the partition wall thickness and $\bar{\delta_{th}}$ or *Nu*. Clearly, *Nu* increases with the partition wall thickness. The maximum value of *Nu* is obtained at *S*^*^ = 0.05. Comparison of the *Nu* of *S*^*^ = 0.01,0.05 shows that the heat flux increases with the partition wall thickness.

To study the influence of the n-type thermal boundary layer distribution on the thermal boundary layer and *Nu*, the gap length is increased (*D*^*^ = 0.020). Figure 12 displays the numerical simulation results for different partition wall thicknesses. The figure shows that the thermal boundary layer distribution for different partition wall thicknesses within the hot channel is n-type. This further indicates that the increase in the partition wall thickness does not affect the thermal boundary layer distribution type when the gap length is fixed. In addition, the thermal boundary layer thickness is not significantly affected by the increase in the partition wall thickness. Figure 13 displays the effect of partition wall thickness on $\bar{\delta_{th}}$. The $\bar{\delta_{th}}$ remains constant as the partition wall thickness increases. The subfigure of Figure 13 clearly shows that $\bar{\delta_{th}}$ and *Nu* remain constant with increasing partition wall thickness. This indicates that the thermal boundary layer and heat flux are not affected by the increase of the partition wall thickness when the gap length is too large. Comparison of the effects of partition wall thickness on the thermal boundary layer and *Nu*, when *D^*^* = 0.010 and *D^*^* = 0.020, shows that the gap length and partition wall thickness have a coupled effect on the thermal boundary layer and heat flux.

## 4. Conclusions

In this study, the thermal boundary layer with different gap lengths and partition wall thicknesses in partitioned thermal convection were investigated. The following conclusions were obtained.

The definition of thermal boundary layer thickness: the temperature field and streamline distribution of partitioned thermal convection show that the thermal boundary layer has spatial nonuniformity. The previous definition of the thermal boundary layer thickness was extended by analyzing the temperature profile of different bulk regions. The thermal boundary layer thickness is defined as the distance between the extrapolation of the linear part of the temperature profile and the horizontal line passing through the maximum local temperature. Moreover, the time evolution of the thermal boundary layer thickness further proved the spatial nonuniformity of the thermal boundary layer.

The effect of partition on thermal boundary thickness: our statistics of the thermal boundary layer thickness at different positions show that the gap length significantly influences the thermal boundary layer distribution, but the partition wall thickness has no obvious effect on the thermal boundary layer distribution. Furthermore, according to the shape of the hot channel distribution, the thermal boundary layer can be divided into two types: m-type and n-type. The two types of thermal boundary layer distribution are mainly affected by the gap length. The thermal boundary layer distribution is m-type when $D^{*}<0.0150$ and is n-type when $D^{*}\geq 0.0150$. 

The effect of partition on heat flux: the $\bar{\delta_{th}}$ and heat flux were comprehensively analyzed. The results show that $\bar{\delta_{th}}$ and heat flux significantly increase with decreasing gap length. The maximum heat flux is obtained when *D*^*^ = 0.0050. The gap length and partition wall thickness have a coupled effect on the thermal boundary layer and heat flux. When *D*^*^ = 0.0100, the increase in the partition wall thickness significantly affects $\bar{\delta_{th}}$ and the heat flux. When *D*^*^ = 0.0200, the increase in the partition wall thickness has no obvious influence on $\bar{\delta_{th}}$ and the heat flux.

## Figures and Tables

**Figure 1 entropy-25-00386-f001:**
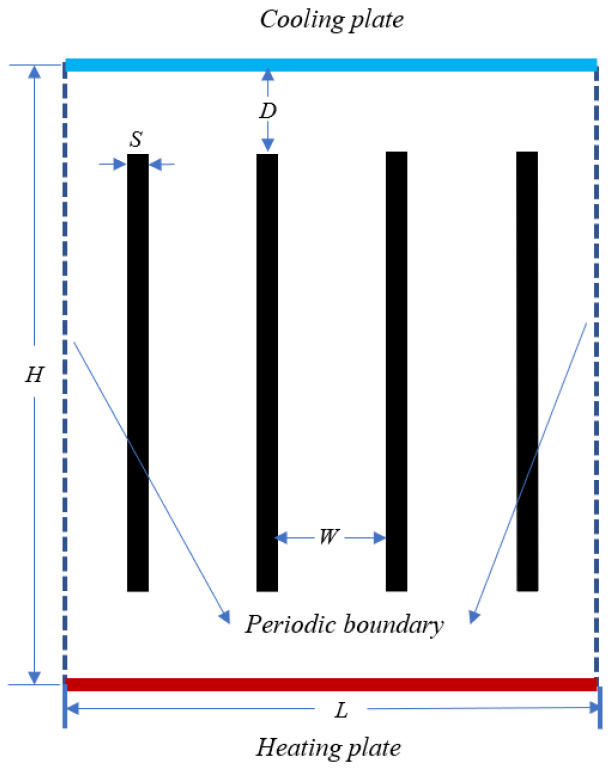
Schematic of the computational domain and boundary conditions.

**Figure 2 entropy-25-00386-f002:**
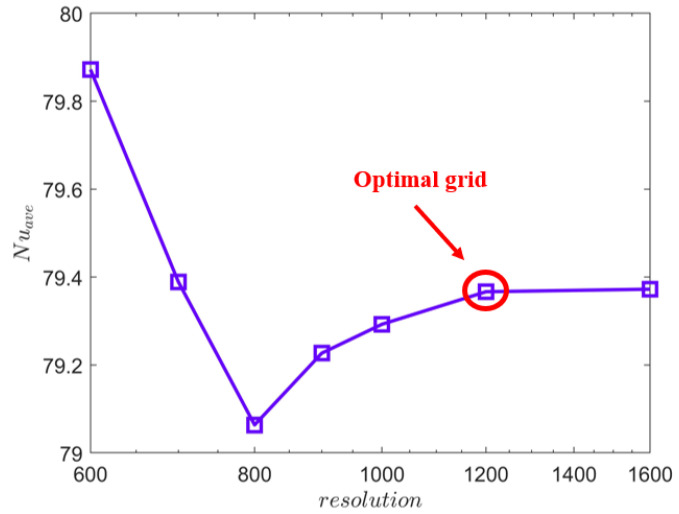
Grid independence test of the average *Nu* at different resolutions.

**Figure 3 entropy-25-00386-f003:**
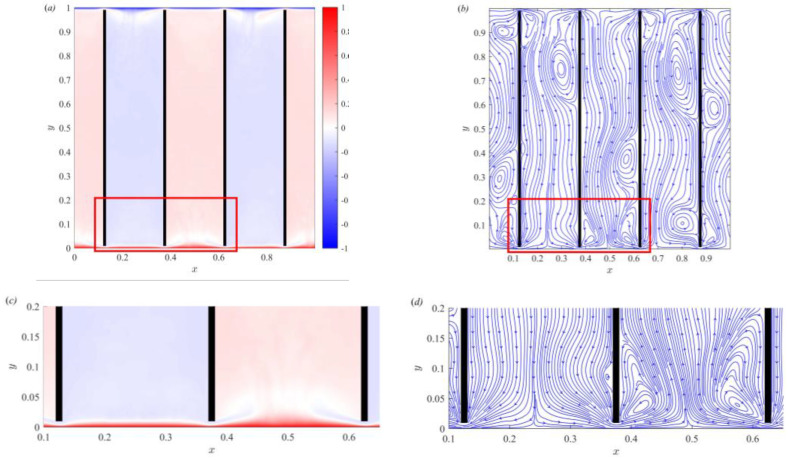
Time-averaged temperature field and streamline distributions of global (**a**,**b**) and local (**c**,**d**). The red boxs in (**a**,**b**) correspond to the enlarged region of (**c**,**d**), respectively.

**Figure 4 entropy-25-00386-f004:**
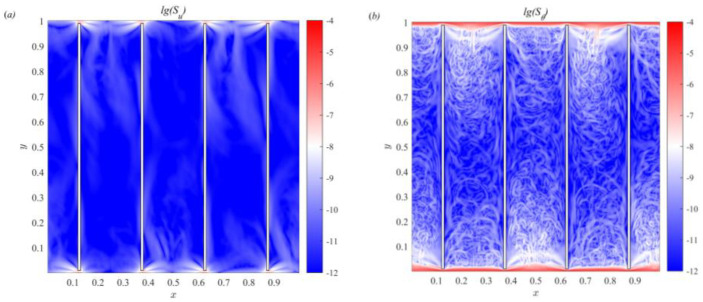
Time-averaged logarithmic viscous (**a**) and thermal (**b**) entropy generation rates fields.

**Figure 5 entropy-25-00386-f005:**
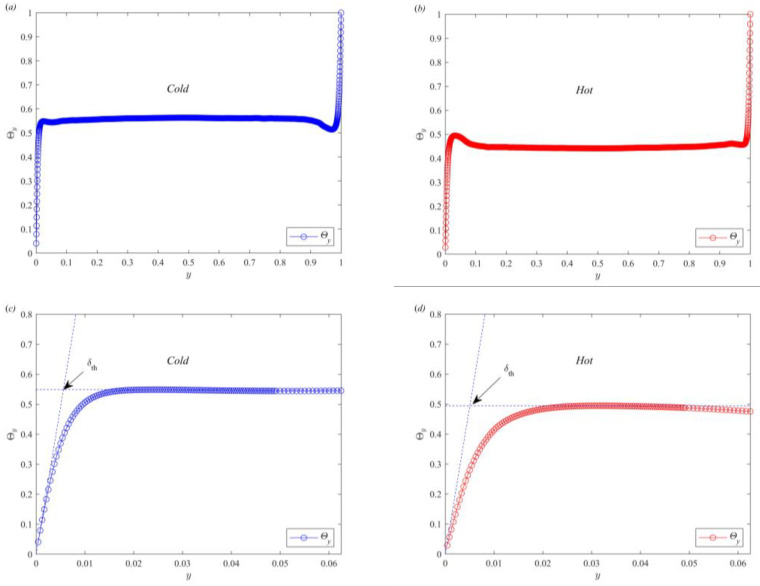
Temperature profile of cold (**a**,**c**) and hot (**b**,**d**) channels.

**Figure 6 entropy-25-00386-f006:**
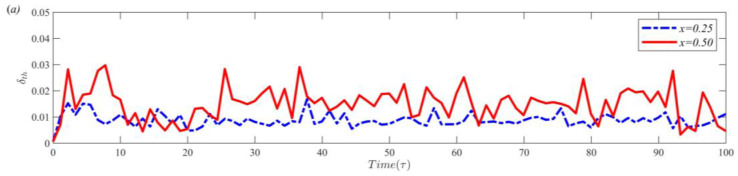
Time evolution of $\delta_{th}$ at different positions (**a**) and regions (**b**).

**Figure 7 entropy-25-00386-f007:**
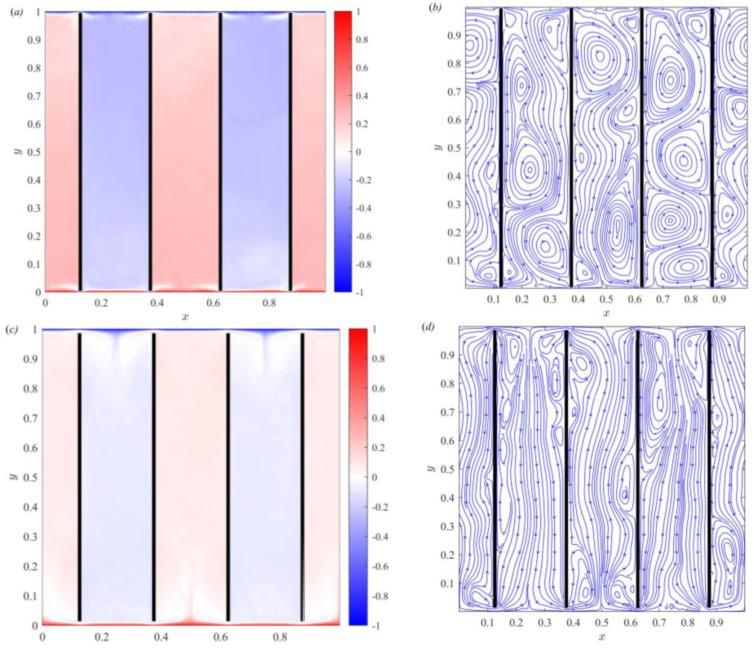
Time-averaged temperature field and streamline distributions of (**a**,**b**) *D*^*^= 0.005, (**c**,**d**) *D*^*^= 0.015 and (**e**,**f**) *D*^*^= 0.030.

**Figure 8 entropy-25-00386-f008:**
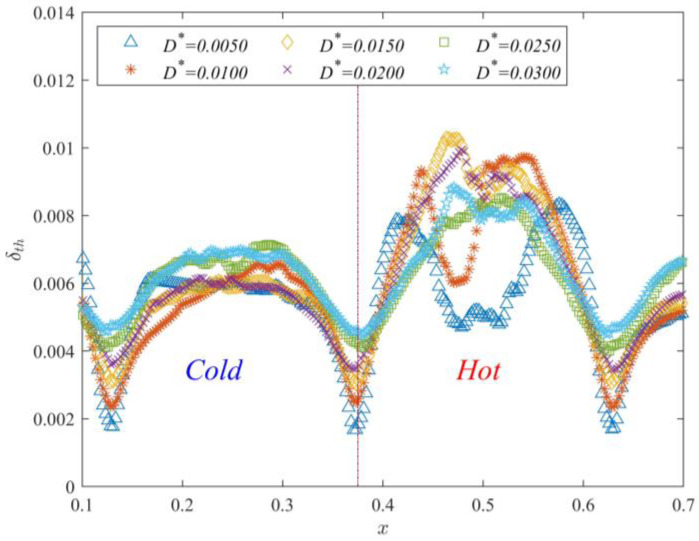
Time-averaged $\delta_{th}$ distribution of different *D**.

**Figure 9 entropy-25-00386-f009:**
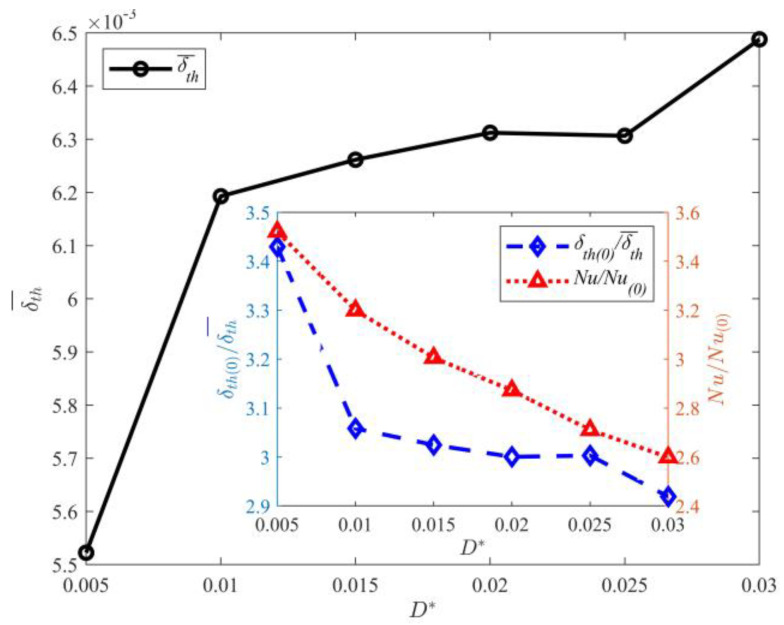
Effect of gap length on $\bar{\delta_{th}}$.

**Figure 10 entropy-25-00386-f010:**
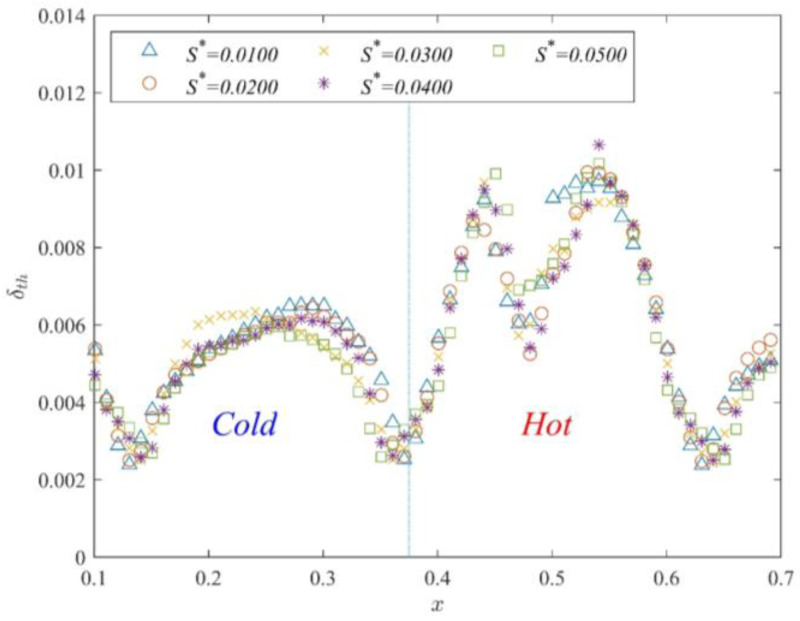
Time-averaged $\delta_{th}$ distribution of different *S*^*^ at *D*^*^ = 0.010.

**Figure 11 entropy-25-00386-f011:**
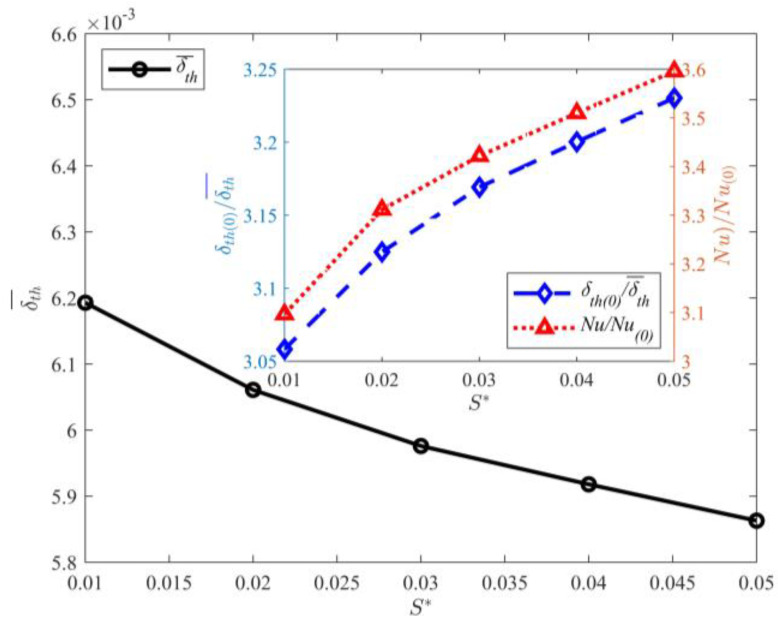
Effect of partition wall thickness on $\bar{\delta_{th}}$ at *D*^*^ = 0.010.

**Figure 12 entropy-25-00386-f012:**
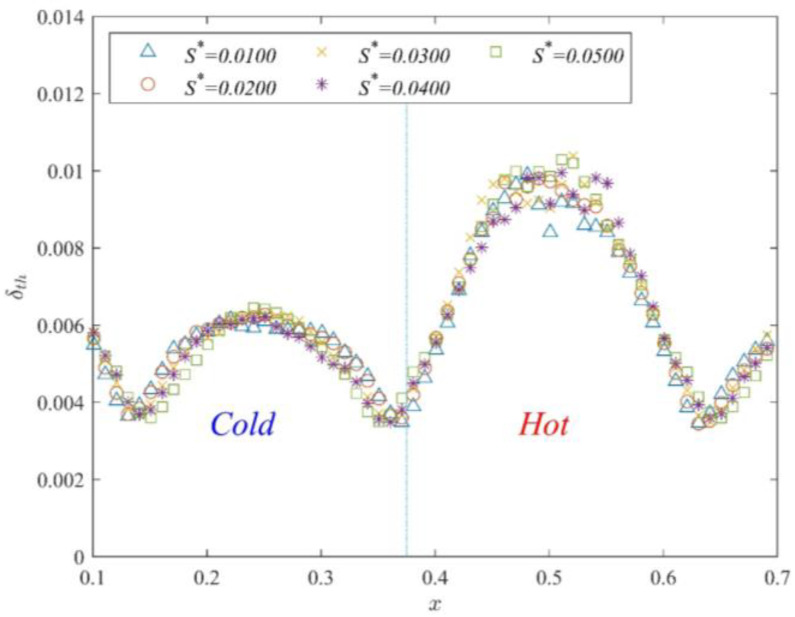
Time-averaged $\delta_{th}$ distribution of different *S*^*^ at *D*^*^ = 0.020.

**Figure 13 entropy-25-00386-f013:**
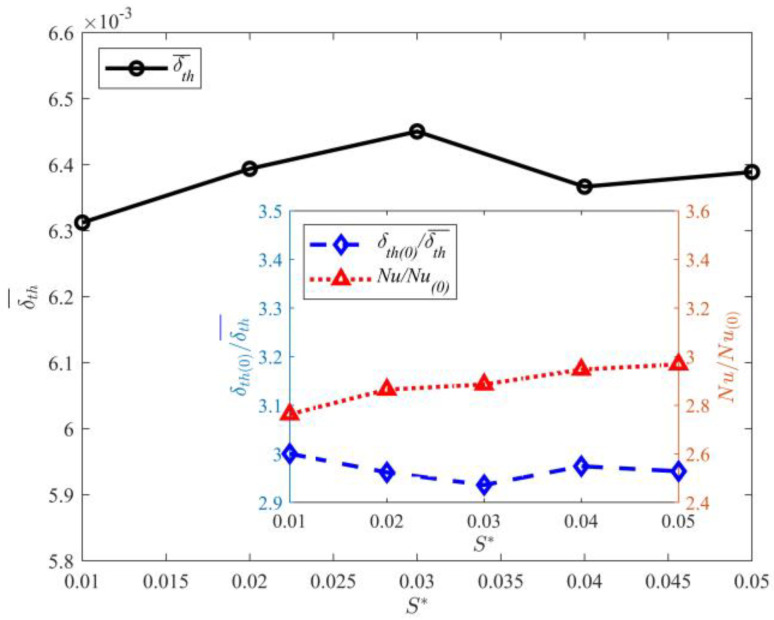
Effect of partition wall thickness on $\bar{\delta_{th}}$ at *D*^*^ = 0.020.

**Table 1 entropy-25-00386-t001:** Comparison of the *Nu* with previous studies.

References	*Ra*	*Pr*	*Nu*
Bao et al. [18]	1 × 10^8^	4.3	25.85
Zhou et al. [37]	1 × 10^8^	4.3	25.62
Present	1 × 10^8^	4.3	25.78

## Data Availability

The data that support the findings of this study are available from the corresponding author upon reasonable request.

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
