# Peer review of "Effect of Gap Length and Partition Thickness on Thermal Boundary Layer in Thermal Convection"

_entropy, 2023, doi:10.3390/e25020386_

Round 1
Reviewer 1 Report
Based on Lattice Boltzmann method (LBM), the authors have evaluated natural convection heat transfer in an enclosure in the presence of the partitions. The enclosure was saturated via pure Water. Effect of the partitions on the boundary layer was probed.
General speaking, the manuscript has not enough innovation. In addition, the manuscript has a major defect in its body. Despite of the partitions have a considerable thickness, thermal (energy) equation in the solid walls has not been introduced and coupled with the others. The overall quality of the study is weak and it needs a significant effort to be improved.
1. What is the proposed novelty of the present study? Presence of the partitions has already been evaluated in the different studies. Providing a clarification is necessary.
2. Consideration the partitions brings "Conjugate natural convection" via thick walls; however, it has not been taken into the account in the present study, since the authors have not considered the related equation in the solid walls.
3. There is not any comparison between present code and the results obtained in the other valid papers. Indeed, without validation the obtained results cannot be considered as correct. The model and code validation and verification are essential steps of a numerical study.
4. Around the grid independency process, it seems the probed grid sizes are out of range. So, please add a view of the selected grid size to the manuscript.
5. Density of the symbols in figures 9 and 11 is very high. Please reduce number of the symbols in the mentioned figures.
6. About the conclusion section, it had better to provide the most important results in the case/ point format.
7. Title of the manuscript is too general. Please consider details contributions in the title.
8. Please consider a nomenclature list and define all the parameters, variables and abbreviations appeared in the manuscript or make sure all symbols have been introduced in first use.
9. Introduction section is not a common. It should include reviewing the related papers from different aspects. The demonstrated descriptions are not enough and the most part of the reviewed papers are accumulated. Such contribution cannot be acceptable.
Author Response
We thank the reviewer for the valuable suggestions and hope that the revised manuscript addresses the concerns raised adequately. The responses are in the attached files.

Reviewer 2 Report
The present reviewer read the MS and believe it can be published after some revisions:
1. It is recommended to add a validation section to valid the present results with the previous ones.
2. Please add a nomenclatures to make the MS more readable.
3. What is novelty of the present work? Please explain at the end of introduction section.
4. Introduction section must be polished, has been written poorly. Some related papers recommended to address in the introduction section:
https://doi.org/10.1016/j.physa.2018.09.164
https://doi.org/10.1016/j.ijheatmasstransfer.2018.05.064
5. Please explain the value of the Δx and Δt. Also the value of the Cs.
The main question is the effect of partition locations on the temperature field.
The method is not new and used several times before, but the geometry is new.
Methodology is OK.
Introduction section (references) must be improved as suggested in comments.
Author Response
We thank the reviewer for the valuable suggestions and hope that the revised manuscript addresses the concerns raised adequately. Our responses are in the attached file.

Reviewer 3 Report
Effect of partitions on thermal boundary layer in thermal convection by Wang et al.
This manuscript studies the Rayleigh Bérnard convection in a partitioned domain where the bottom plate is heated and the top plate is cooled. They perform 2D numerical simulations of this phenomenon using the Lattice Boltzmann method. They validate their numerical code before presenting the results with earlier studies and also by doing grid independence tests. So, their results are well validated. I also enjoyed reading the results sections where the authors perform a thorough analysis of the thermal boundary layer and study the effects of various parameters especially the gap thickness on the boundary layer. Overall, the results are interesting and the manuscript is well written, I would be happy to recommend it for publication with minor revisions. See below.
Minor:
The authors perhaps can discuss the practical importance of the geometry that they are using in the paper. Yes, we know that RBC is important in natural and practical applications, but in what applications does partitioned geometry is encountered? Also, authors can try to use their findings to explain how the understandings they develop will be useful in these applications.
line 48 – Bao et al. – give reference number and year of publication
Line 84 – give long form for LBM before using the abbreviation

Author Response

(The authors gave the same response as above.)

Reviewer 4 Report
In this manuscript, the authors adopt the lattice Boltzmann method as the tool for direct numerical simulation of turbulent thermal convection. Particularly, the effect of inserting the gap on the thermal boundary layer thickness is discussed. The paper is well-written, and the results are reliable. The work will attract those working on turbulent convection. I am glad to recommend the publication of this manuscript in Entropy. Below are some minor comments/suggestions for the authors to further improve the paper.
1. Page 3, Line 85: The authors describe the advantages of the LB including “excellent parallelism”. Here, the corresponding reference to support this statement should be given. Examples include but are not limited to LBM parallel computing on GPU platforms, such as DOI: 10.1016/j.ijheatmasstransfer.2017.02.032 and DOI: 10.1016/j.ijheatmasstransfer.2022.123649.
2. I appreciate the authors used local refinement to carefully resolve the boundary layers while saving computational cost in the bulk region. This technique is useful in the current problem. In Eq. 20, it would be helpful for the readers to know more details if the authors can give values of n (representing the ratio of the coarse grid to the fine grid).
3, Page 12, Line 353: The authors claim that the thermal boundary layers distribution with varying gap length can be divided into two types: one is M type, and the other should be explicitly termed. In addition, the authors may discuss the possible physical mechanism.
Author Response

(The authors gave the same response as above.)

Round 2
Reviewer 1 Report
The authors have satisfactorily addressed my concerns and the revised manuscript is of better quality than the original submission. Therefore, the manuscript can be recommended for publication.
Reviewer 2 Report
The authors addressed all the comments.
Reviewer 3 Report
I am happy to recommend the manuscript for publication.
Reviewer 4 Report
The authors have improved the paper, and I am glad to represent the paper published in the present form.